# Classification of commercial districts based on predicting the survival rate of food service market in Seoul

DongHyeon Lee[1], Jaekyung Lee[2], ManSu Kang[3], SangHyun Cheon[2]*

1 Centre for Small Business Insight, Seoul Credit Guarantee Foundation, Seoul, Republic of Korea,
2 Urban Design & Planning, Hongik University, Seoul, Republic of Korea, 3 Korea Data Bank Inc., Seoul, Republic of Korea.

☯ These authors contributed equally to this work.
* scheon@gmail.com

## Abstract

The chronic small-business closure has emerged as a critical economic issue in South Korea, considering more than half of businesses have been closed within three years. While some previous literature has analyzed the causes and their negative impacts on the economy, there is still a lack of studies on understanding the pattern dynamics and predicting future possible closure scenarios due to the lack of appropriate data. Using 3,000,000 individual commercial facility data from 2004 to 2018 in Seoul, Korea, the primary purpose of this research is two-fold: (1) to develop a methodological framework to simulate survival rate pattern change using a deep-learning based model and (2) to reclassify the commercial districts based on the prediction outcomes to 8 survival rate change types. The results indicate that the LSTM model can be useful in predicting and simulating the survival rate of commercial facilities. Moreover, the CBD area showed a decrease in the survival rate in the future, and the commercial districts around university districts and IT industry clusters were divided into commercial districts with an increased survival rate in the future. The results of this study are expected to be used as quantitative evidence for more direct and realistic policy establishment.

## 1. Introduction

Commercial districts are places where the city's social and economic activities occur. In the commercial area, urban residents consume various goods and services, which becomes a source of income for local small business owners. Preventing the decline of the commercial district and inducing its revitalization enables the continuous supply of goods and services to city residents. In addition, it is possible to prevent a decrease in the value of the city by increasing tourism income through the influx of outsiders. Among diverse activities in the commercial district, Small local

**Data availability statement:** According to the internal regulations of the Seoul Credit Guarantee Foundation under the Seoul Metropolitan Government, readers cannot access the analyzed data. This restriction is in place to protect the personal information of small business owners. The Seoul Credit Guarantee Foundation is the entity imposing this restriction. Researchers interested in accessing this dataset may submit an application to the Seoul Credit Guarantee Foundation to request access. Please contact the Data Access Officer at the Seoul Credit Guarantee Foundation (e-mail: sdatadam@seoulshinbo.co.kr).

**Funding:** This work was supported by the National Research Foundation of Korea grants (No. 2020R1A2C2008443) and (No. 2023R1A2C2003030) funded by the Korea government (MSIT) (SH.C.). This work was also supported by 2023 Hongik University Research Fund (SH.C.).

**Competing interests:** The authors have declared that no competing interests exist.

business(mainly self-employed) is one of the most critical activities that can sustain the local economy by bringing out local characteristics and attracting people.

Small local businesses accounted for 25% of all South Korean industries in 2017, ranking Korea fifth out of 35 OECD nations [1]. As the Baby Boomer generation retires and young people increasingly start enterprises due to difficulty finding jobs, competition among small local businesses has become fierce. Meanwhile, the economic climate is deteriorating, with business closures rising due to increased minimum wage and decreased consumption as the economy stagnates. Under the circumstances, the closure rate of firms in 2019 versus those that opened in Korea was over 72 percent, and more than half of enterprises now close within three years after opening [2].

Since the restaurant industry occupies 25.7% of the Small local business in the commercial districts, it can be an indicator for measuring the degree of activation of commercial districts [3]. Due to the low barrier to entry, the restaurant industry accounts for the highest among all small local business openings and closures. In the fourth quarter of 2019, small business owners in Seoul showed 31,141 opening and closing(opening: 15,286, closing: 15,855), of which the restaurant industry accounted for 12,868 opening and closing(opening: 6,775, closing: 6,093), the highest at 41% [3](S1 Fig). And the average operating period of the restaurant industry has been decreasing from 3.1 years to 2.5 years in the past 5 years [3] (S1 Fig). Because of the high level of closures in the restaurant industry, there has been an increase in the unstable employment climate and massive economic losses for both individuals and the country.

Furthermore, nationwide events also influenced the survival and closure of small businesses. During the study period from 2004 to 2018, notable events included the 2008 global financial crisis originating in the United States, the widespread adoption of regional currency starting in the mid-2010s, and the 2015 MERS outbreak. Among these, the introduction of regional currency positively impacted small businesses by redirecting consumer spending away from large retail chains and promoting local consumption [4]. In contrast, the global financial crisis and the MERS outbreak negatively affected small businesses by significantly reducing overall consumer spending [5,6].

Unfortunately, many small local businesses are still indiscriminately open due to the lack of an accurate understanding of current market conditions and prediction of future commercial changes. Since various variables are complexly related to the sales of small local businesses, it is necessary to build a large amount of data to understand the market condition. Furthermore, it is required to use accurate prediction models based on machine learning to predict future commercial changes.

Thus, this study builds an LSTM-based restaurant industry survival rate prediction model based on over 2.94 million retail stores' opening and closing business data in Seoul. Then, we reclassify the commercial district types using k-means Clustering based on the durability dynamics prediction results. Since various business pattern dynamics respond sensitively to internal and external factors, it is critical to developing a more accurate prediction model to understand the business trends and forecast future possible pattern changes to establish more realistic policies for small business owners in the region.

## 2. Literature review

### 2.1. Survival of food service business

In 2021, the closure rate of food service businesses compared to the opening was 89.8%, and more than half of the stores will close within three years [7]. The frequent closure of small businesses entails economic losses to the state and individuals, such as job instability and increased social costs [8]. Accordingly, to maintain a more stable survival rate for small business and reduce the vacancy rate, various studies have been conducted on the closure of small businesses and related factors influencing it.

The frequent closure of restaurants has long been the focus of researchers [9]. Researchers measured the closure rate and survival period of the restaurant industry. They compared them with other sectors or compared and analyzed them by group according to region, industry, and employee size [10–14]. Furthermore, it was revealed that the success and closure of the restaurant industry could be influenced by not only a store's individual characteristics, such as location, size, provided products, and financial assets, but also road density, land price, and land use [12,15–17]. Several studies have conducted survival analysis and multinomial logistic regression analysis for restaurants located in Seoul, Korea, to influence the business period of restaurants, such as resident population, floating population, mixed-use, number of similar businesses, affiliation, transportation accessibility, and plot size [8,18].

As such, many studies have derived factors affecting businesses' business continuity and survival rate and analyzed the business period using statistical models. However, since most of the studies identified problems at a specific point in time through past data or analyzed their impact, there were limitations in predicting future commercial district changes based on the research results and presenting policies that can be applied to them. Therefore, this study intends to design a survival rate prediction model based on the business closure impact factors derived from existing studies and predict the business's future survival rate.

### 2.2. Prediction model based on neural networks

With the development of computer technology and federal data organizations since the 1950s, statistics-based models for analyzing and predicting various economic and environmental phenomena in cities have emerged, and multiple studies have demonstrated that the statistical models contribute to predicting various urban phenomena. However, since the traditional regression-based models have limitations in reflecting the non-linear trend of the market, neural network-based prediction models have risen in popularity to predict spatial pattern dynamics [19–22].

Youn and Gu predicted the closure rate of Korean lodging companies through logistic regression and artificial neural networks and compared the accuracy of the two models [19]. As a result, the accuracy of the neural network model was higher than that of the logistic regression model, confirming that the neural network model was suitable for predicting the closure rate. Furthermore, Limsombunchai and Temur et al. also proved the neural network model has higher performance in house price forecasting than traditional regression models [20,21]. Lee et al. predicted the vacancy rate of commercial buildings in Seoul through the LSTM model and analyzed the impact of individual building variables, location variables, and local economic variables on the vacancy rate [22]. This study also showed that the LSTM model effectively predicts the future of commercial districts.

Several studies have demonstrated that the predictive power of deep learning models is superior to that of regression models, especially those utilizing LSTM in time series analysis. However, since most studies are limited to specific areas, such as real estate prices and traffic forecasting, this study intends to confirm the applicability of the prediction model based on deep learning in commercial area analysis.

### 2.3. Classification of commercial districts

The commercial areas that revitalize the local economy and the city through exchanges of various resources and services are critical areas of the urban space [23,24]. Many studies have attempted to define the commercial district and analyze

the characteristics of each commercial district. Using Bayesian classification, Araldi and Fusco classified commercial areas in Liberia, France, into eighteen types, depending on the distribution of stores and the type of industry [25].

Pinedo and Gutierrez divided Madrid into 600m*600m pixels and classified them into ten types through k-means Clustering using the number of buildings, total floor area, number of businesses, and social network data variables [26]. Colaco and Silva classified commercial areas in Lisbon into commercial district types based on the density and diversity of stores through k-means Clustering [27]. After categorizing the commercial district types for three-time points in 1995, 2005, and 2010, they observed the change of the commercial district type according to time frame. Kim and Choi classified Seoul into four types using the increase and decrease in business establishments and closures from 2005 to 2010 [28]. Jung classified Seoul into six types based on the temporal and spatial patterns of changes in sales [29]. Lee et al. divided the commercial districts of Seoul by reflecting changes in all six variables, such as sales, office worker population, floating population, household income, and the number of businesses, and compared the survival period of stores located in each commercial district [2].

Although several studies have been conducted to reclassify commercial district types and developmental stages, most have focused on the current state of the commercial district or changes from the past to the present. There are no studies that predict changes and use them to classify types.

## 2.4. Literature gaps

Although multiple studies have been conducted to identify business durability and understand both causes and consequences of commercial activity, this research differs from previous research on the survival rate of small businesses and commercial activity in three ways.

First, this study developed the deep learning-based prediction model to forecast restaurant business durability, which accounts for the largest percentage of self-employed businesses in Korea due to low entry barriers. While deep-learning prediction models have shown superior performance in terms of accuracy compared to conventional statistical models in the real estate field, this method has not yet been used in commercial analysis. Thus, we utilized the long short-term memory (LSTM) model to predict the survival rate changes stores more accurately with many influences and fluctuations over time.

Second, this study utilized 3,000,000 individual commercial facility data from the Seoul Credit Guarantee Foundation, including location, sales, rent, and opening and closing dates. Due to a lack of data, previous studies failed to consider various variables affecting business continuity and apply them to a whole city level [8,12,16–18]. Using the commercial district-related spatial big data, we predicted the survival rate of restaurants in Seoul in units of a 300m*300m pixel.

Third, this study reclassified commercial districts based on future possible survival rate pattern dynamics. Existing studies have classified commercial districts based on past changes. However, small stores, which account for most self-employed businesses, change rapidly over time and must reflect future changes to establish more effective anti-closing measures. Therefore, this study classifies the types of commercial districts based on predicted changes in future survival rates of stores and analyzes their characteristics to provide evidence for self-employed business prevention.

Therefore, this research's primary purposes are (1) to develop a methodological framework to simulate survival rate pattern change using a deep-learning based model and (2) to reclassify the commercial districts based on the prediction outcomes to 8 survival rate change types. Considering the circumstance where the closure rate of the food service industry is increasing yearly, it is critical to understand future survival rate changes and establish evidence-based strategies for each commercial district using the accurate prediction output.

## 3. Methods

The City of Seoul in Korea is utilized as the study area, where more than 500,000 stores in Seoul and over 4,000 open and close monthly, causing active commercial district dynamics [3]. Firstly, to predict future possible survival rate pattern

dynamics (between 2019 and 2023), approximately 2,940,000 individual stores' opening and closure data from 2004 to 2018 were collected by Seoul Credit Guarantee Foundation. Considering the area of Seoul and computer performance, we set up a 300 m x 300m grid (total of 7,108 pixels) as the analysis unit. After dividing the entire area into 7,108 pixels and excluding pixels with no commercial data due to topography (river, mountain, vacant property, military base), we used 4,034 grids to predict future survival rates. We reclassified commercial districts (Fig 1).

This study predicts the survival rate of restaurant businesses in Seoul by developing a LSTM prediction model based on a cyclic neural network. And we also suggest future policy directions according to the characteristics of Seoul's commercial district types. The research process can be into two parts(Fig 2): (1) predicting the survival rate of the restaurant and (2) classifying the types of commercial districts.

[1] To predict the survival rate of the restaurant industry using the LSTM model: first, we selected the independent variables of the survival rate prediction model in consideration of the previous studies that studied factors affecting the business durability of stores. Second, we constructed data for three-time intervals of the selected variables for 2004–2008, 2009–2013, and 2014–2018. Third, we estimated the survival rate from 2019 to 2023 through the LSTM model trained on data from 2004 to 2018.

[2] To classify the types of commercial districts, the 4,034 pixels were divided into three categories based on the number of businesses per pixel: low, medium, and high-density commercial districts. Then, we derived pixel clusters by performing k-means Clustering using time-series changes in survival rates for each commercial district. Third, we classified the commercial district types of each pixel cluster using a confidence interval. Lastly, the spatial distribution of each type of commercial district was confirmed to suggest future policy directions according to the characteristics of the commercial districts.

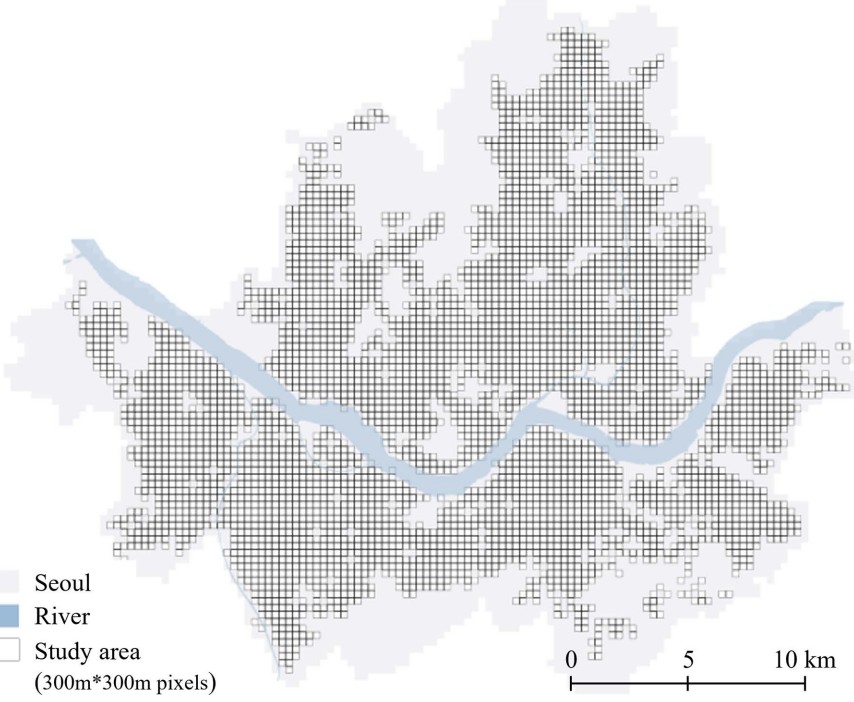

**Fig 1. Study area.**

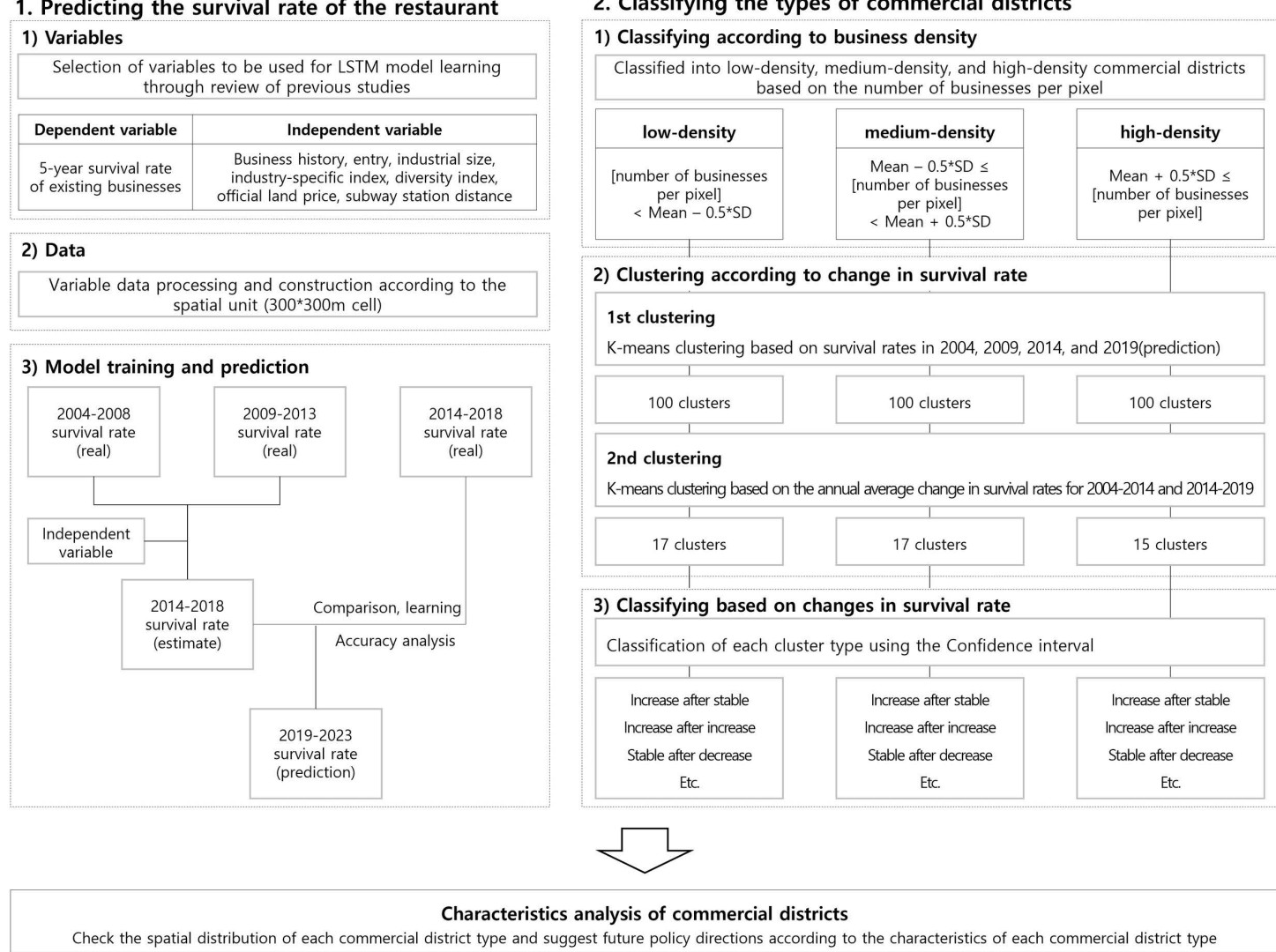

**Fig 2. Research Diagram showing predicting process (LSTM) & classifying the types of commercial districts.**

### 3.1. Predicting the survival rate of the restaurant using LSTM

**3.1.1. Variables.** Since variable selection can significantly affect prediction outcomes, the causal mechanism related to survival rate is identified based on the literature review. Seven different driving factors were chosen considering the data capability and literature(Table 1). The dependent variable is the survival rate of the stores, which is the probability of how many stores operating in year t will survive in year t + 5 at the pixel. In South Korea, the average operational period for small businesses is 5.6 years [30]. Therefore, we determined that a 5-year interval would be sufficient to capture meaningful changes in business survival. Additionally, the Seoul Credit Guarantee Foundation, which compiles small business statistics for Seoul, observes survival rates for both 1-year and 5-year periods. Furthermore, since previous literature has also used 5-year survival rates to track the business performance of small businesses and to develop policies aimed at improving their sustainability, we also use 5-year survival rates [31,32].

**Table 1. Types and descriptions of variables to predict survival rate.**

| | Variable | Description | Reference |
|---|---|---|---|
| **Dependent variable** | The survival rate of existing businesses | The probability that a business operating in year t in that cell will remain open in year t+5 | |
| **Independent variable** | Business history | The average length of business for businesses located in that cell | [11] [16] [14] |
| | Entry | Number of new restaurants in the cell | [34] [35] [28] |
| | Industrial size | Number of stores in the same industry as the business | [34] [12] [8] [26] |
| | Industry-specific index | The maximum value among the LQ values by industry in the cell compared to Seoul | [33] [27] |
| | Diversity index | The reciprocal of the sum of the difference in the proportion of each sector in the cell compared to Seoul | [33] [27] |
| | Official land price | The average official land price of the cell | [15] [17] |
| | Subway station distance | Distance from the nearest subway station in the cell | [11] [16] [18] |

The independent variables are seven: business history, entry, industrial size, industry-specific index, diversity index, official land price, and distance from a subway station. After arranging the variables that have been shown to affect the business durability and closure of stores in previous studies, we selected variables that can obtain data and set them as independent variables. The definitions and references are described in Table 1. Previous studies have commonly used additional variables such as industry types, number of employees, region, and economic growth rate. Since we focused on predicting survival rates for a single industry (i.e., food industry), the industry types variable was excluded. The number of employees variable was also excluded due to difficulties in data collection. Region and economic growth rate variables are typically used in studies targeting larger areas, such as countries. For a relatively smaller area like Seoul, these two variables were found to be less significant and were therefore excluded from the analysis.

Among the variables used as independent variables, the industry-specific index, indicating the cell's relative degree of industry specialization compared to the whole of Seoul, was derived as the maximum value of the location quotient(LQ) distribution of each industry in the cell compared to Seoul. The diversity index, indicating the relative degree of industry diversity in the cell compared to the whole of Seoul, was derived by taking a reciprocal sum of the differences in the proportion of each sector in the cell compared to Seoul [33]. The calculation formula is as follows Eq (1–2).

The industry classification used in this study to calculate the industry-specific index and diversity index is based on the "One hundred industries closely tied to daily life". In 2019, the Seoul Metropolitan Government defined the "One Hundred Industries Closely Tied to Daily Life" to facilitate statistical analysis and policy support for small local businesses. These industries typically have fewer than five employees and a relatively high ease of entry and exit, and they consist of 10 food service industries, 47 service industries, and 43 retail industries [3]. We choose this classification to create industry specific index and diversity index because it is well-suited for categorizing the businesses in Seoul, which is the study's target area, at a comparable level.

$$RZI_i = Max_j \left( \frac{s_{ij}}{s_j} \right)$$

(1)

$$RDI_i = \frac{1}{\sum_j |s_{ij} - s_j|}$$

(2)

$RZI_i$: The industry-specific index of i cell.
$RZI_i$: The diversity index of i cell.
$s_{ij}$: Ratio of industry j in i cell.
$s_j$: Ratio of industry j in Seoul.

**3.1.2. LSTM model.** An artificial neural network (ANN) belonging to a supervised learning algorithm among machine learning, a field of artificial intelligence, is created by mimicking the human brain's neurons [36]. Several deep-learning based models have been developed over the past 50 years, such as a convolutional neural network mainly used for image processing and a recurrent neural network suitable for time-series data processing, and advanced models are continuously being developed recently.

RNN, one of the deep-learning algorithms using artificial neural networks, has been used in various fields to predict diverse spatial and social pattern dynamics, such as stock price forecasting and real estate price forecasting [37–39]. However, traditional RNN has a problem: the longer the neural network structure, the faster the learning outcomes of the distant past are lost [40]. To complement this, the LSTM model was introduced by Sepp Hochreiter in 1997 [41]. The LSTM model solves the problem in RNNs by adding another layer called Cell State to the hidden layer to determine whether or not to remember new information [42].

In addition to the LSTM model, we considered several potential alternative models, including AR, MA, ARIMA, RNN, and GRU. Statistical models such as AR, MA, and ARIMA are well-suited for linear data; however, previous studies have highlighted their limitations in handling complex, nonlinear patterns often observed in time-series data, such as the survival rates of small businesses [43]. LSTMs (Long Short-Term Memory Networks) and GRUs (Gated Recurrent Units) are both types of recurrent neural networks (RNNs) designed to address the vanishing gradient problem and capture long-term dependencies in sequential data. Due to its simplified structure, consisting of only two gates—the reset gate and the update gate—the GRU exhibits greater computational efficiency compared to the LSTM. This efficiency and a reduced parameter count make GRUs particularly advantageous for scenarios with limited computational resources. In contrast, LSTMs can capture long-term dependencies in sequential data effectively and accurately with three gates (input, forget, and output gates). Since predicting business survival rates can be influenced by various internal/external factors and directly impacts the city's economy, the LSTM was chosen to get more accurate outcomes [44].

In this study, we constructed data from 2004–2008, 2009–2013, and 2014–2018 for model learning and predicted the survival rate of Seoul's restaurant industry businesses from 2019 to 2023. First, we use the survival rates from 2004–2008 and 2009–2013 as input patterns, along with seven independent variables as input factors, to predict the survival rates for 2014–2018. Second, we compare the predicted survival rates for 2014–2018 with the observed survival rates from the same period to train the model and improve its accuracy. Third, after verifying the model's accuracy through training, we use it to predict the survival rates for 2019–2023 (Fig 3).

To construct the most suitable model, we measured the root mean square error (RMSE) between the actual survival rate and the estimated survival rate every 1,000 training sessions(epochs) and analyzed the trend to determine the most optimal epochs. RMSE is one of the values representing the difference between the estimated value and the actual value of the model. The closer the value is to 0, the smaller the difference between the estimated value and the actual value, and the higher the accuracy of the model [22]. The calculation formula is as follows Eq (3).

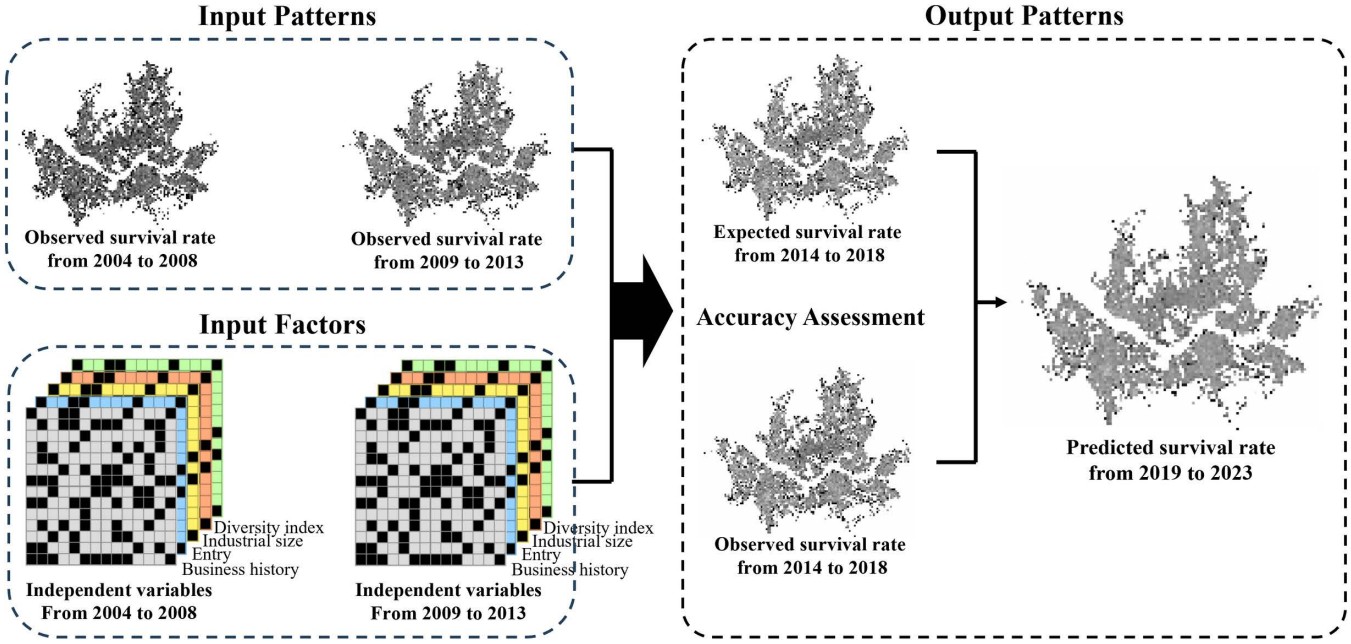

**Fig 3. LSTM conceptual diagram.**

$$RMSE = \sqrt{\frac{\sum_{i=1}^{n} (actual_i - predict_i)^2}{n}}$$

(3)

### 3.2. Classifying the types of commercial districts

**3.2.1. Classifying commercial districts according to business density.** Since several previous studies proved that the business durability and closing rate are different in commercial districts by business density [2,18]. The 4,034 pixels (300 m x 300 m) were firstly reclassified into three different types by the density of commercial facilities: low-, medium-, and high-density districts in Seoul, which was the subject of analysis, according to business density.

The process of classifying commercial areas according to density is as follows. First, we derived the average number of food service businesses operating in 2004, 2009, and 2014 for each pixel and used this as the number of stores in the pixel. The calculation formula is as follows Eq (4).

$$N_i = \frac{(N_i^{2004} + N_i^{2009} + N_i^{2014})}{3}$$

(4)

Second, we checked the distribution by analyzing the basic statistics of the number of stores per pixel and set the standards for low-density, medium-density, and high-density commercial districts using the average and standard deviation of the number of stores. The criteria for classifying commercial districts according to business density are as follows (Table 2).

**3.2.2. K-means clustering.** We classified pixels using k-means Clustering based on the survival rate of stores from 2004 to 2008, 2009–2013, and 2014–2018, and the survival rate of stores from 2019 to 2023 predicted through the LSTM model in this study. K-means clustering is one of the clustering methods that divides data into several groups with similar

**Table 2. Criteria for classifying commercial districts by business density.**

| Commercial districts | Definition |
|---|---|
| Low-density commercial districts | $N_i < Mean - 0.5SD$ |
| Medium-density commercial districts | $Mean - 0.5SD \leq N_i < Mean + 0.5SD$ |
| High-density commercial districts | $Mean + 0.5SD \leq N_i$ |

characteristics and belongs to the unsupervised learning algorithm of machine learning. The K-means clustering algorithm divides the data to minimize the average Euclidean distance between the center of each cluster and the objects belonging to the cluster. The algorithm is expressed as the following Eq (5).

$$V = \sum_{i=1}^{k} \sum_{j \in S_i} |x_i - m_i|$$

(5)

V: the overall variance
Si: the set of points belonging to the cluster
mi: the average of the individuals in each cluster

First, we classified the 300m pixels in each commercial area into 10–20 clusters through k-means Clustering, which variables the survival rate of existing businesses in four-time intervals (2004–2008, 2009–2013, 2014–2018, and 2019–2023). It was difficult to derive the characteristics of the cluster due to various changes in the survival rates of pixels belonging to a cluster. In particular, since the survival rate dynamics in low- and medium-density commercial areas were larger and more rapid than that of high-density commercial districts, there was a limit to classifying the districts with only one stage of k-means Clustering.

Therefore, we classified the types of changes in survival rates in low, medium, and high-density commercial districts through two stages of k-means Clustering. Through the first k-means clustering, we divided pixels in each commercial district into 100 clusters using the survival rate values of four-time intervals as variables. Through several simulations, we confirmed that when the number of clusters was set to 100, pixels showing similar survival rate changes were classified into the same cluster in low, medium, and high-density commercial districts. Then, we derived one trend line based on the survival rate changes of each pixel for every 100 clusters classified through the above process.

Through the second k-means Clustering, we clustered 100 clusters classified in the first k-means Clustering using the 2004–2014 annual average change in survival rate and the 2014–2019 average survival rate change in the trend line as variables. The average annual change in the survival rate from 2004 to 2014 is the difference between the survival rate(2004~2008) and the survival rate(2014~2018) divided by 10, which is the difference between the years of the section and refers to the past change in the survival rate in the food service industry. The average annual change in survival rate from 2004 to 2014 is the difference between the survival rate from 2014 to 2018 and the survival rate predicted in this study for 2019–2023 divided by 5, the year difference between the sections, meaning the future change in the survival rate in the food service industry. The calculation method is as follows Eq (6). AAC in the formula represents the average annual change, and S represents the survival rate.

$$Past\ change : AAC_{2004}^{2014} = \frac{S_{2014}^{2018} - S_{2004}^{2008}}{2014 - 2004}$$

$$Future\ change : AAC_{2014}^{2019} = \frac{S_{2019}^{2023} - S_{2014}^{2018}}{2019 - 2014}$$

(6)

Since secondary k-means clustering is the final step in determining clusters, a more accurate method determined the number of clusters. We calculated the BSS (between the sum of squares)/WSS (within the sum of squares) according to the number of clusters for each commercial district and derived a section in which the value rapidly increased and reflected this to determine the number of clusters.

**3.2.3. Classifying commercial districts based on survival rate dynamics.** To divide the types of clusters into "increasing," "decreasing," and "stable" survival rate changes, we derived a confidence interval for the change in survival rate for each low-, medium-, and high-density commercial district. We used it as a quantitative criterion to classify the types of clusters. The classification process is as follows(Fig 4).

First, set the interval boundaries. We derived a population average of 99% confidence intervals for the existing and future changes in the survival rate of the restaurant industry, centered at 0, and set it as the "stable" zone judgment interval. Then, we divided into groups larger and smaller than the stable zone judgment interval, derived each 99% confidence interval, and set it as the judgment interval for "increasing" and "decreasing" commercial zones.

Second, divide into increasing, decreasing, and stable commercial districts according to the interval boundaries. We divided the clusters on the left, including the decreasing commercial zone judgment section, into the existing decreasing commercial zone (a) and the clusters on the right, including the increasing commercial zone judgment section into the existing increasing commercial zone (c). After that, we divided the clusters between the increasing and decreasing commercial zones, including the stable zone judgment section, into the existing stable zone (b). Since then, we divided the future increase, decrease, and stable zone in the same way for the future change rate (1, 2, 3).

Third, we classified the types of commercial districts into nine classes: increase after decrease (IaD; a, 1), increase after stable (IaS; b, 1), increase after increase (IaI; c, 1), stable after decrease (SaD; a, 2), stable after stable (SaS; b, 2), stable after increase (SaI; c, 2), decrease after decrease (DaD; a, 3), decrease after stable (DaS; b, 3), and decrease after increase (DaI; c, 3).

After classifying the types of commercial districts through the above method, we identified the spatial distribution of each type of commercial district and suggested the direction of self-employment policy suitable for the characteristics.

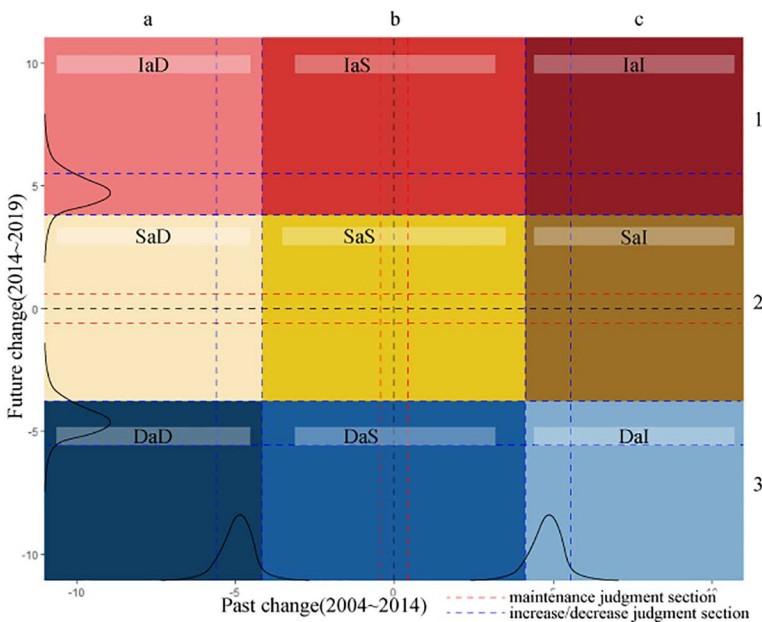

**Fig 4. Classifying commercial districts based on survival rate changes.**

## 4. Results

### 4.1. Predicting the survival rate of the restaurant

**4.1.1. Basic statistics of dependent and independent variables.** Before predicting the future survival rate of existing restaurant businesses, we analyzed the basic statistics of variables used to build the LSTM model(Table 3).

While the average survival rate (the probability that a business operating in t years will be operating in t+5) decreased over time to 57.42% in 2004, 50.35% in 2009, and 44.02% in 2014, the average operating period of a business within one pixel, increased sharply from 40.62 months in 2004 to 63.32 months in 2009 and 80.93 months in 2014. Furthermore, the number of newly established businesses (the entry variable) has gradually increased from 2004 to 2014. The official land price increased steeply, and the distance to subway stations gradually decreased due to the establishment of subway stations. The results indicate that the sustainability of the restaurant industry decreases. This might be due to the early closure of new businesses rather than the closure of businesses operating for a long time.

**4.1.2. Prediction of the LSTM model.** The LSTM model in this study estimates the survival rate for 2014–2018 based on the survival rates of existing restaurants in 2004–2008 and 2009–2013 and the data of seven independent variables that affect them. After verifying the model's accuracy through comparative analysis with the actual survival rate from 2014 to 2018, the model estimates the expected survival rate from 2019 to 2023. Fig 5 indicates the training process of each epoch with RMSE. RMSE was derived for every 1,000 training cycles to validate the model performance. As a result, RMSE, initially 29.79 in the 10,000th learning, fell to 1.29, meaning the accuracy tended to increase as the number of

**Table 3. Basic statistics of dependent and independent variables.**

| Variable | Time Interval | N | Mean | SD | Median | Min | Max |
|---|---|---|---|---|---|---|---|
| Survival rate(%) | 2004-2008 | 4034 | 57.42 | 25.97 | 60.00 | 0.00 | 100.00 |
| | 2009-2013 | 4034 | 50.35 | 22.99 | 50.00 | 0.00 | 100.00 |
| | 2014-2018 | 4034 | 44.02 | 22.85 | 43.75 | 0.00 | 100.00 |
| Business history | 2004-2008 | 4034 | 40.62 | 33.69 | 37.50 | 0.00 | 1142.00 |
| | 2009-2013 | 4034 | 63.32 | 37.88 | 60.54 | 0.00 | 1202.00 |
| | 2014-2018 | 4034 | 80.93 | 40.62 | 78.45 | 0.00 | 654.00 |
| Entry | 2004-2008 | 4034 | 31.13 | 39.24 | 17.00 | 0.00 | 368.00 |
| | 2009-2013 | 4034 | 30.72 | 40.34 | 16.00 | 0.00 | 485.00 |
| | 2014-2018 | 4034 | 32.59 | 43.10 | 18.00 | 0.00 | 511.00 |
| Industrial size | 2004-2008 | 4034 | 59.47 | 76.06 | 32.00 | 0.00 | 831.00 |
| | 2009-2013 | 4034 | 67.94 | 86.67 | 37.00 | 0.00 | 934.00 |
| | 2014-2018 | 4034 | 69.29 | 88.02 | 38.50 | 0.00 | 932.00 |
| Diversity index | 2004-2008 | 4034 | 1.04 | 0.37 | 0.99 | 0.00 | 2.40 |
| | 2009-2013 | 4034 | 1.08 | 0.37 | 1.04 | 0.00 | 2.39 |
| | 2014-2018 | 4034 | 1.07 | 0.36 | 1.05 | 0.50 | 2.23 |
| Industry-specific index | 2004-2008 | 4034 | 31.14 | 49.45 | 17.91 | 0.00 | 1448.80 |
| | 2009-2013 | 4034 | 26.62 | 39.20 | 15.69 | 0.00 | 912.93 |
| | 2014-2018 | 4034 | 25.45 | 35.63 | 15.27 | 3.45 | 555.58 |
| Official land price | 2004-2008 | 4034 | 2.83E+08 | 5.45E+08 | 1.81E+08 | 0.00 | 1.97E+10 |
| | 2009-2013 | 4034 | 3.66E+08 | 6.41E+08 | 2.32E+08 | 0.00 | 1.75E+10 |
| | 2014-2018 | 4034 | 4.36E+08 | 7.33E+08 | 2.8E+08 | 0.00 | 1.84E+10 |
| Subway station distance | 2004-2008 | 4034 | 702.59 | 558.79 | 549.50 | 11.42 | 4836.25 |
| | 2009-2013 | 4034 | 630.29 | 474.07 | 510.88 | 11.42 | 3868.98 |
| | 2014-2018 | 4034 | 627.42 | 474.49 | 506.84 | 11.42 | 3868.98 |

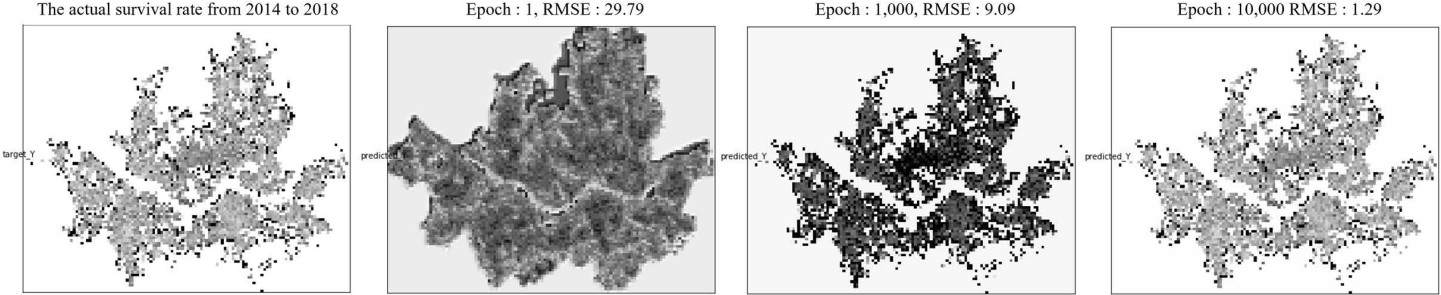

| The actual survival rate from 2014 to 2018 | Epoch : 1, RMSE : 29.79 | Epoch : 1,000, RMSE : 9.09 | Epoch : 10,000 RMSE : 1.29 |

**Fig 5. LSTM training process with RMSE by epochs.**

training rose. Since the RMSE did not decrease significantly after learning 10,000 times, we set the number of times to train the model to 10,000 times, considering the overfitting problem.

The results of predicting the survival rate of existing businesses in 2019–2023 using the learned model are shown in Fig 6 below. The average survival rate of the food service industry from 2019 to 2023 was predicted to be 41.56%. By region, the future survival rate was predicted to be low in (1)Gangnam, (2)Guro-dong, and (3)Cheonho-dong, and the future survival rate in (4)Seoul Station, (5)Hongdae, and (6)Banghak-dong was predicted to be high. A more detailed analysis result using the predicted survival rate is described in the next section.

### 4.2. Classifying the types of commercial districts

**4.2.1. Classifying commercial districts according to business density.** As a result of analyzing the basic statistics of the number of businesses in 4,034 pixels in Seoul, the average number of businesses per pixel is 34.09, and the standard deviation is 43.82. According to the classification criteria described in Table 2 above, we define a commercial district with 12.18 or fewer businesses in each pixel as a low-density commercial district, a commercial district with more than 12.18 and 55.99 or less as a medium-density commercial district, and a commercial district with more than 55.99 as a high-density commercial district. There were 1,668 pixels in low-density commercial districts, 1,525 in medium-density commercial districts, and 841 in high-density commercial districts.

Low-density commercial districts are in suburban areas and regions with high-density residential facilities. Medium-density commercial districts are found around the high-density commercial districts and neighborhood commercial areas near residential areas. High-density commercial districts are located in well-known commercial areas of Seoul, central business districts, amusement park vicinities, and several university neighborhoods.

Over the 15-year analysis period of this study, business densities within each pixel may have changed, potentially leading to shifts in density types. To examine the temporal stability of commercial district classifications based on business density, we classified density types separately for three time periods (2004–2008, 2009–2013, and 2014–2019) and identified patterns of density changes and their distributions. For example, if a pixel was classified as a low-density commercial district in 2004–2008, a medium-density commercial district in 2009–2013, and a high-density commercial district in 2014–2019, it was categorized as an LMH type.

Approximately 90% of pixels in low-density commercial districts remained classified as low-density across all three periods, indicating stability over time(Fig 7-A). In the medium- and high-density commercial districts, 29% and 27% of pixels, respectively, showed changes. This higher variability compared to low-density districts may reflect the challenges of maintaining medium- or high-density commercial districts in real-world conditions(Fig 7–B, C). However, most of the pixels that changed classifications were either categorized in the same district type for two consecutive time periods with increasing density levels (e.g., LMM and MMH types in medium-density districts, and MHH types in high-density districts)

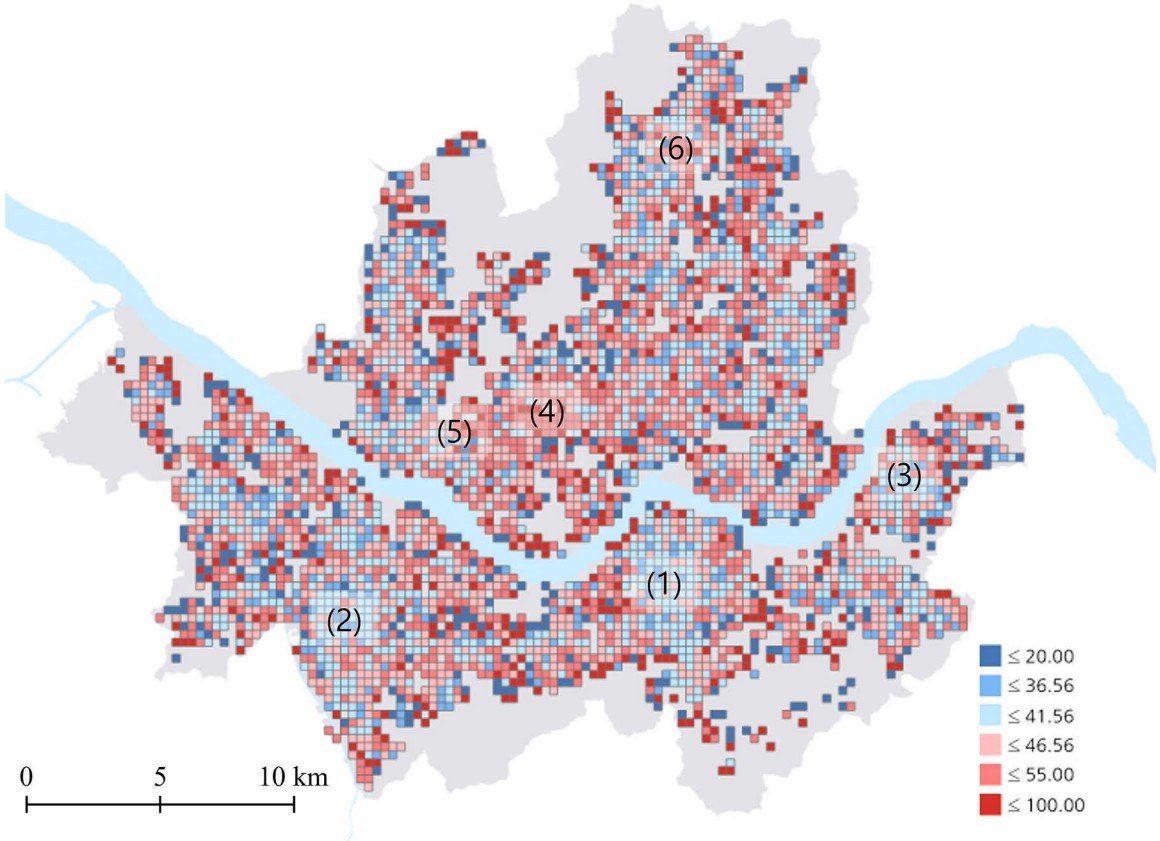

| | |
|---|---|
| | ≤ 20.00 |
| | ≤ 36.56 |
| | ≤ 41.56 |
| | ≤ 46.56 |
| | ≤ 55.00 |
| | ≤ 100.00 |

**Fig 6. Survival rate prediction of restaurants in 2019-2023 using the learned model.**

or transitioned to a higher-density district and later returned to the original type (e.g., MHM type in medium-density districts). Other types of transitions were relatively rare, and their spatial distributions did not significantly differ from the general characteristics of their respective density categories.

Therefore, this study's method of classifying density types based on the average business density across the three time periods was valid. Furthermore, we confirmed that the spatial distribution of high-density commercial districts classified in this study closely aligns with the areas defined as developed commercial districts by the Seoul Metropolitan Government, which are characterized by high density and active commercial activity(Fig 7–C, D). This further supports the validity of the density classification method used in this study to reflect real-world conditions.

Table 4 shows the basic statistics of the survival rate of the restaurant for each low-, medium-, and high-density commercial district. In all three commercial districts, the average survival rate decreased over time. The standard deviation was the largest in the low-density commercial district and the smallest in the high-density commercial district, and the range was the widest in the low-density commercial district and the narrowest in the high-density commercial district.

**4.2.2. Clustering according to change in survival rate.** As described in the method of the previous study, we clustered 4,034 pixels in Seoul by performing k-means Clustering twice based on time-series changes in the survival rate of existing businesses in the restaurant industry.

First, we divided low-density, medium-density, and high-density commercial districts into 100 clusters through k-means Clustering and derived a trend line of the change in survival rate for each cluster. Afterward, for the second k-means Clustering, we derived the BSS/WSS according to the number of clusters and determined the appropriate number of

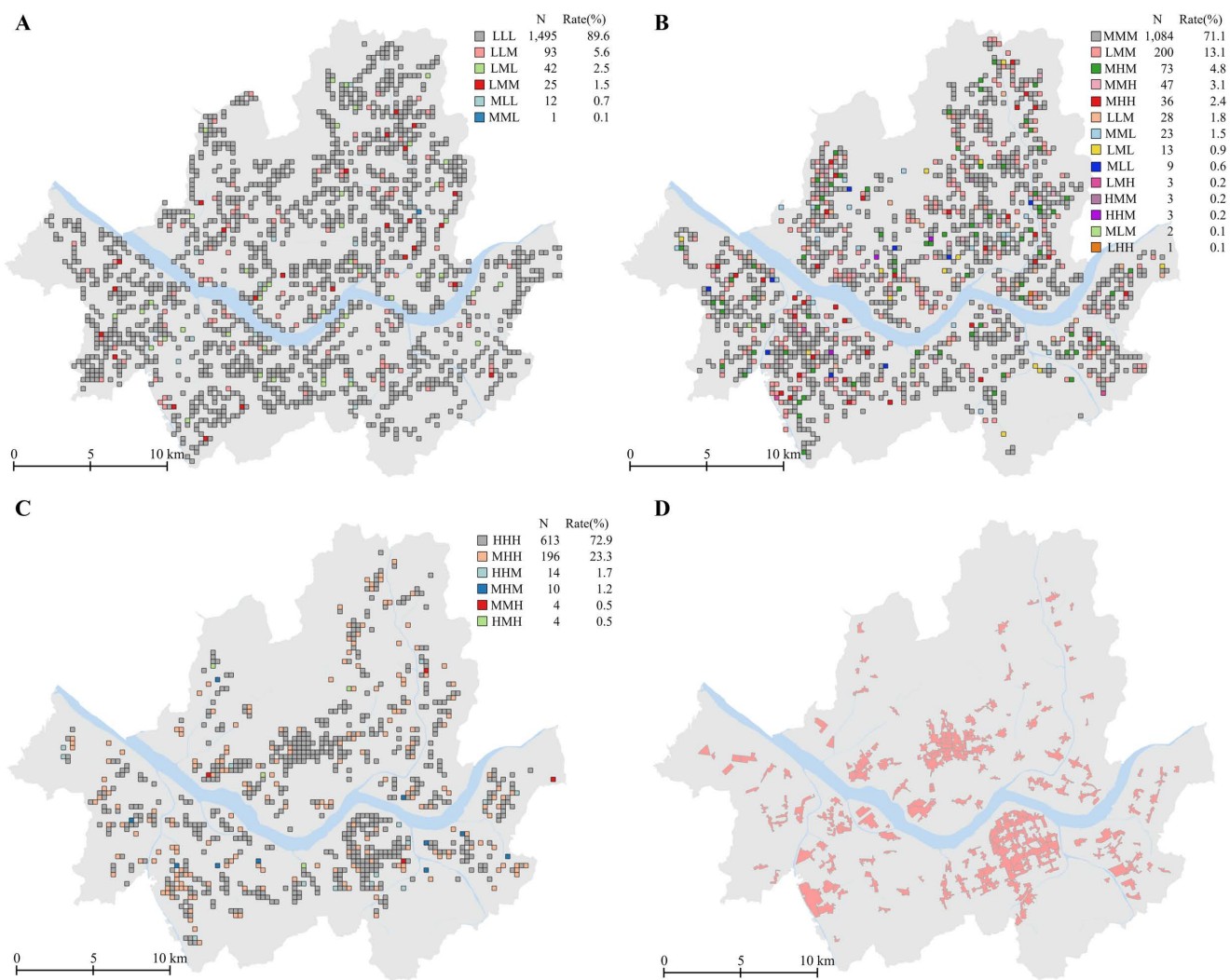

**Fig 7. Classifying commercial districts by business density.** (A) Low-density commercial districts. (B) Medium-density commercial districts. (C) High-density commercial districts. (D) Developed commercial districts. Source: Seoul Metropolitan Government [30].

**Table 4. Basic statistics of the survival rate.**

|  | Time Interval | N | Mean | SD | Median | Min | Max |
|---|---|---|---|---|---|---|---|
| Low-density commercial district | 2004-2008 | 1668 | 52.92 | 37.63 | 50.00 | 0.00 | 100.00 |
|  | 2009-2013 | 1668 | 49.11 | 33.86 | 50.00 | 0.00 | 100.00 |
|  | 2014-2018 | 1668 | 44.48 | 33.45 | 46.15 | 0.00 | 100.00 |
|  | 2019-2023 | 1668 | 38.89 | 24.79 | 44.05 | 0.00 | 100.00 |
| Medium-density commercial district | 2004-2008 | 1525 | 60.74 | 12.87 | 60.00 | 11.11 | 100.00 |
|  | 2009-2013 | 1525 | 51.52 | 10.79 | 51.22 | 15.79 | 100.00 |
|  | 2014-2018 | 1525 | 43.76 | 11.16 | 43.64 | 5.88 | 100.00 |
|  | 2019-2023 | 1525 | 43.74 | 7.58 | 43.40 | 0.00 | 95.35 |
| High-density commercial district | 2004-2008 | 841 | 60.34 | 7.70 | 60.29 | 35.71 | 100.00 |
|  | 2009-2013 | 841 | 50.67 | 6.73 | 50.00 | 32.88 | 79.31 |
|  | 2014-2018 | 841 | 43.58 | 7.76 | 43.14 | 14.89 | 72.73 |
|  | 2019-2023 | 841 | 42.92 | 5.08 | 42.55 | 15.53 | 64.46 |

clusters for each commercial district. As a result, the sections where the BSS/WSS value rises rapidly are 16–17 sections, 34–35 sections in the low-density commercial district, 16–17, 22–23 sections in the medium-density commercial district, and 14–15, 18–19, 22~23 sections in the high-density commercial district. Using a smaller number, we decided to classify low-density and medium-density commercial districts into 17 clusters and high-density commercial districts into 15 clusters (Fig 8).

The secondary k-means clustering results shown on the coordinate plane are shown in Fig 8 below. At the bottom of Fig 8, the X-axis means the average annual change in survival rate from 2004 to 2014(past change), and the Y-axis means the average annual change in the survival rate from 2014 to 2019(future change). In the case of low-density commercial districts, the distribution of change in the survival rate was the widest. As many clusters appear in the second and fourth quadrants, there are many pixels with an increase in the survival rate in 2004–2014 and a decrease in the survival rate in 2014–2019. In the case of medium- and high-density commercial districts, the distribution of changes in survival rates was relatively narrow. As most of the clusters appear in the second and third quadrants, the survival rate of most of the pixels in the medium and high-density commercial areas decreased from 2004 to 2014.

**4.2.3. Classifying commercial districts based on changes in survival rate.** As described in Section 3.2.3, we classified the commercial district types of clusters by using the confidence interval of the past and future changes in the survival rate of the restaurant. The number of pixels for each commercial district type is shown in Table 5 below. In the case of the low-density commercial district, out of 1,668 pixels, the stable after stable commercial district overwhelmingly had the most with 630 pixels, followed by the stable after decrease commercial district with 316 and the decrease after stable commercial district with 208. In the case of the medium-density commercial districts (total of 1,525 pixels), the stable after stable commercial district had the most with 367 pixels, the stable after decrease commercial district with 343, and the decrease after stable commercial district with 298 pixels. In the case of the high-density commercial districts, "IaS", "SaI", and "DaI" commercial districts did not appear, so they were classified into five types of commercial districts.

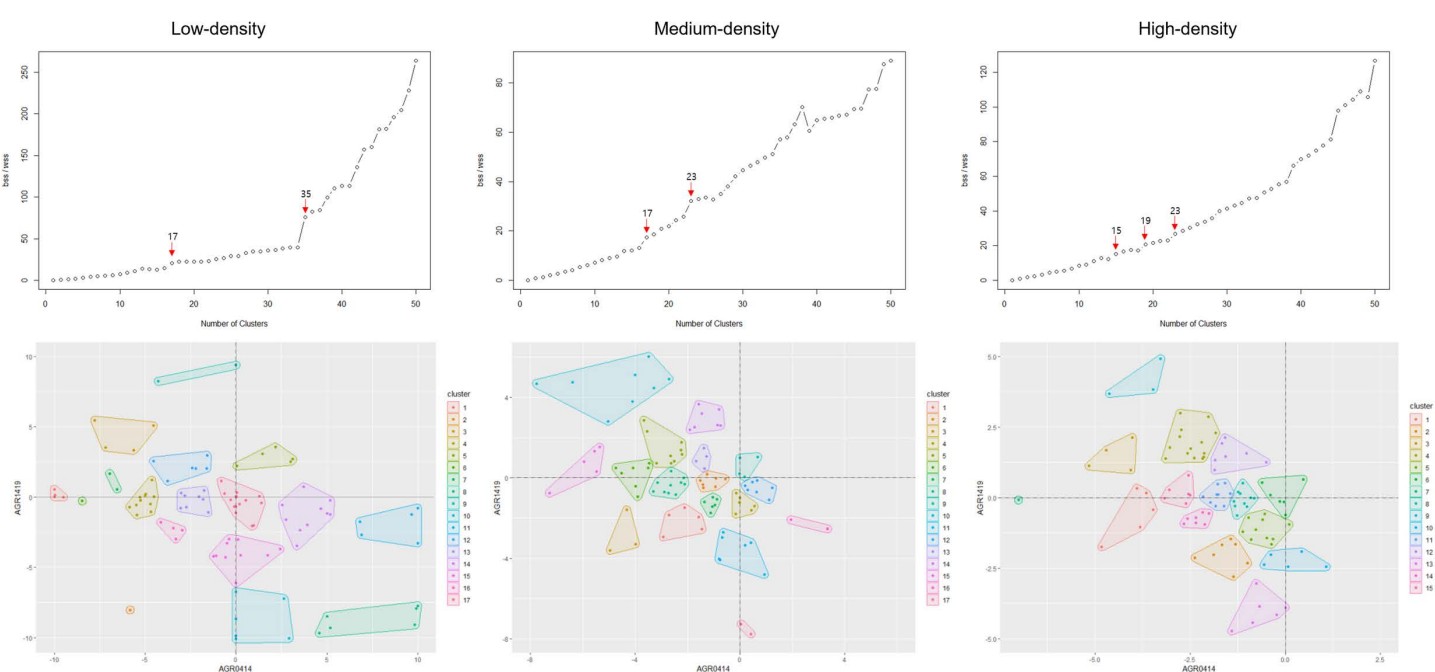

**Fig 8. Determination of the number of clusters and clustering results.**

**Table 5. The number of pixels for each commercial district type.**

| Commercial district type | Low-density | Medium-density | High-density | Sum |
|---|---|---|---|---|
| IaS: Increase after stable | 71 | 67 | | 138 |
| IaD: Increase after decrease | 163 | 267 | 165 | 595 |
| SaI: stable after increase | 176 | 77 | | 253 |
| SaS: stable after stable | 630 | 367 | 165 | 1,162 |
| SaD: stable after decrease | 316 | 343 | 319 | 978 |
| DaI: decrease after increase | 102 | 14 | | 116 |
| DaS: decrease after stable | 208 | 298 | 133 | 639 |
| DaD: decrease after decrease | 2 | 92 | 59 | 153 |
| Sum | 1,668 (41.35%) | 1,525 (37.80%) | 841 (20.85%) | 4,034 |

Stable after decrease commercial district accounted for the largest number with 319, followed by a decrease after decrease commercial district and stable after stable commercial district with the same number of 165 pixels.

The spatial distribution of each type of commercial district is as follows in Fig 9. Most low-density commercial districts with less than 12.18 businesses per pixel are located in residential or outlying areas with low density, making it difficult to define the location of pixels as a specific area.

The areas known as representative commercial areas of Seoul are the Central Business District (Jongro-gu, Jung-gu), University Town area (Hongdae, Sinchon), Gangnam Business District (GBD; Gangnam-gu, Seocho-gu), and IT Business District (Guro-gu). First, most of the CBD (Fig 10-A) were commercial districts that declined in the future, such as "DaS" and "DaD," and some "IaD" commercial districts appeared. This is presumed to be because there are mainly old commercial districts in the CBD, and local revitalization of commercial districts is taking place.

The campus town area around the university district (Fig. 10-B) had a decreased survival rate in the past, but in the future, the survival rate will increase or be stable in many cases. Due to the constant demand for young people in their 20s and 30s, the commercial district continuously expands into the surrounding area.

In the case of GBD around the business area where major companies are located (Fig. 10-C), various types of commercial districts were mixed, such as "SaD," "DaS," "DaD," and "IaD." In addition to many offices, various urban facilities, such as academies and lodging facilities, are concentrated in this area, so the commercial district is also very diverse.

In the IT Business District around workplaces where IT companies are concentrated (Fig 10-D), the main commercial districts were "SaS" and "IaD." In particular, the type of "IaD" was prominent in the commercial districts around the business area and the foreign residence (Chinese) groups. This seems to be attributable to increased demand due to the growth of IT companies and increased foreign immigration.

To understand the types of changes in the survival rate of major commercial districts in Seoul, we reclassified 253 developed commercial districts and 1,010 alley commercial districts in Seoul provided by the Seoul Commercial District Analysis Service based on the commercial district type results of this study. For a more concise classification, among the eight types of commercial districts, we define "IaS" commercial districts as increased commercial districts, "DaS," "DaD" commercial districts as decreased commercial districts, and "SaD," "SaS," "SaI" commercial districts as stable commercial districts. Accordingly, we reclassified the 8 commercial district types into 5 (Table 6).

Decrease, maintenance, and "IaD" commercial districts were the most common, and stable commercial districts appeared the most among them. There were a few increased commercial districts and a decrease after increase commercial districts. The decreased commercial district and the increase after decrease commercial districts were similar in the alley commercial district. In contrast, in the developed commercial district, the decrease in commercial district was more than twice as much as the increase after decrease commercial district. It was found that there is a strong tendency for the survival rate of restaurants to decrease in the developed commercial district with a high density of stores.

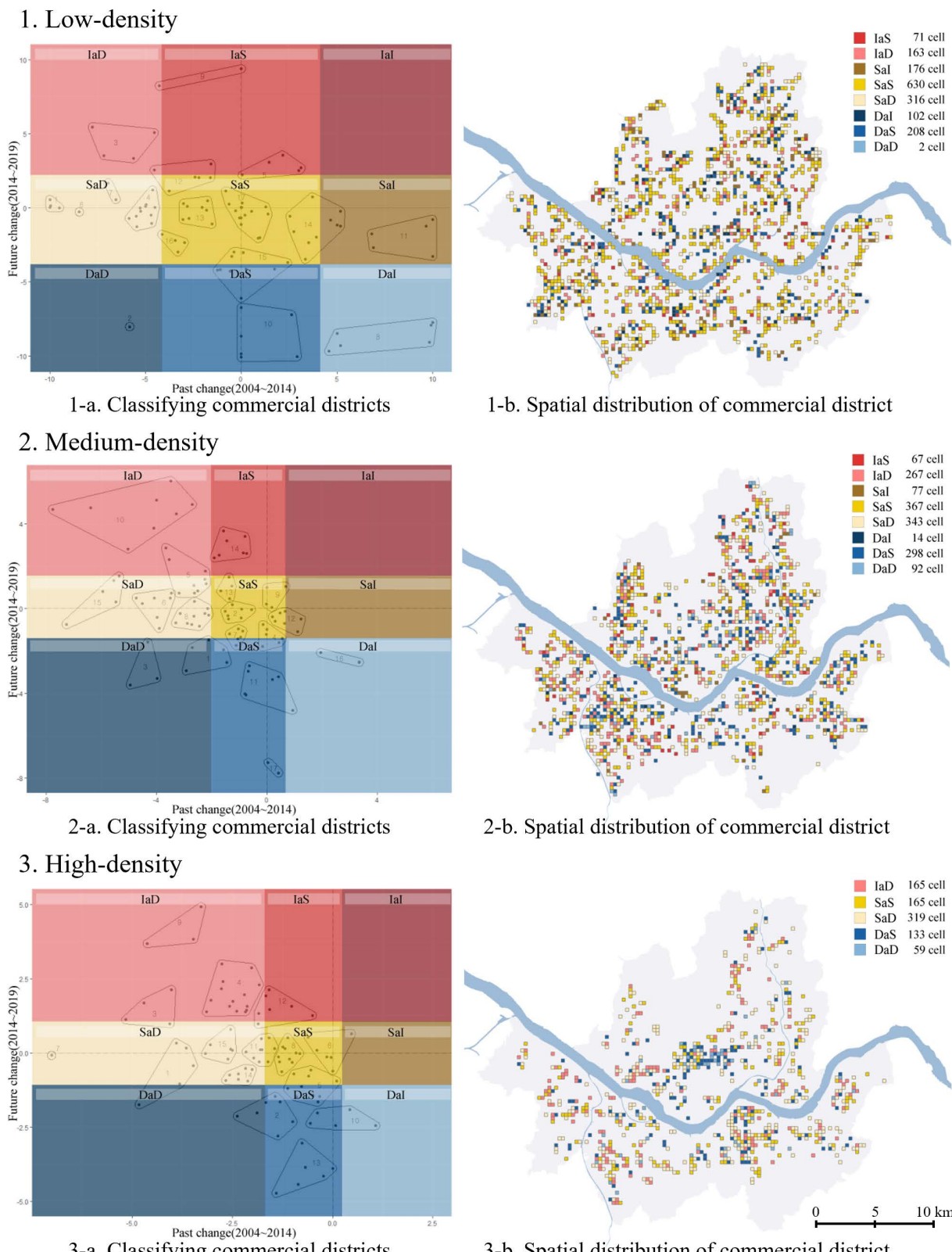

**Fig 9. Classifying commercial districts and spatial distribution.**

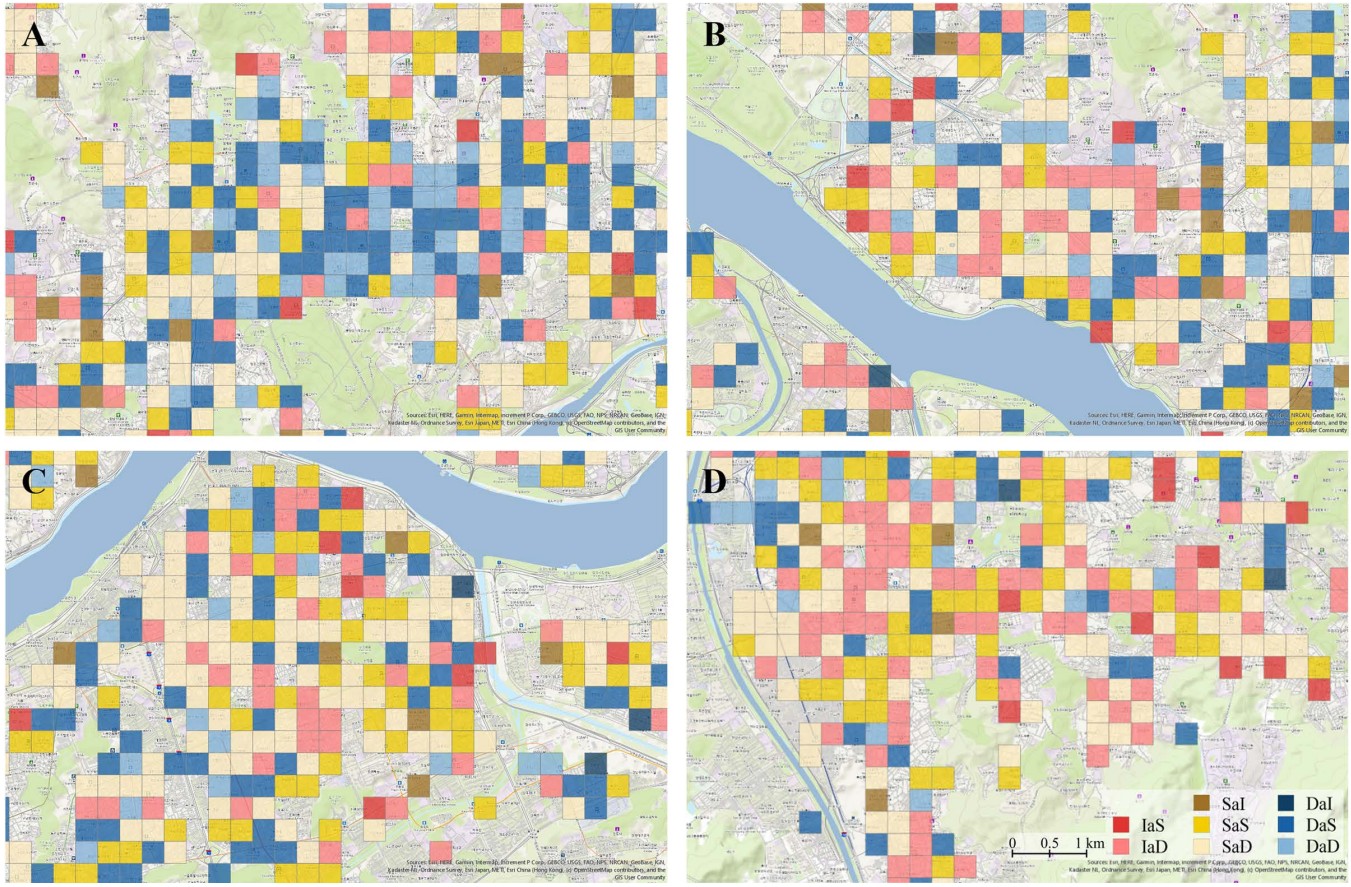

**Fig 10. Classification outcomes of four major commercial districts.** (A) Classifying in CBD. (B) Classifying in campus town area. (C) Classifying in GBD. (D) Classifying in IT Business District.

**Table 6. Classification of developed/alley commercial district.**

| Commercial district type | All | | Alley commercial district | | Developed commercial district | |
|---|---|---|---|---|---|---|
| Increase | 24 | (1.9%) | 23 | (2.3%) | 1 | (1.8%) |
| Decrease | 256 | (21.2%) | 199 | (19.7%) | 57 | (26.1%) |
| Stable | 755 | (59.3%) | 584 | (57.8%) | 171 | (67.6%) |
| DAI | 4 | (0.3%) | 4 | (0.4%) | 0 | (0.0%) |
| Increase after Decrease | 224 | (17.4%) | 200 | (19.8%) | 24 | (9.5%) |
| Sum | 1263 | (100.0%) | 1010 | (100.0%) | 253 | (100.0%) |

## 5. Discussion

This study aims to predict the survival rate of existing restaurants from 2019 to 2023 based on data from about 3 million restaurants in Seoul and to reclassify the types of commercial districts based on time-series changes in survival rates. Based on previous studies, we found that most studies related to commercial district analysis drew limitations that analyzed commercial district changes at a specific point in the past by relying on statistical techniques for reasons such as lack of data. Traditional statistical models, such as ARIMA and GRU models, were considered but found

to be limited in handling the complex, nonlinear patterns inherent in time series data. To overcome the limitations of these preceding studies, we constructed a deep learning-based LSTM model to predict the 5-year survival rate of restaurants in Seoul. Then, we reclassified the types of commercial districts through k-means Clustering based on the prediction results.

The results of this study are as follows. First, through the accuracy analysis of the LSTM-based predictive model, it was found that the LSTM model developed in this study effectively predicted the survival rate of restaurants. The results revealed that the future survival rate of major downtown areas in Seoul was low. In contrast, the neighborhood commercial areas adjacent to residential or commercial areas around business areas were high. Given its ability to capture complex spatiotemporal patterns, the LSTM-based model holds great potential for application in other domains where dynamic forecasting is required.

Second, as a result of classifying the types of commercial districts in Seoul based on the change in the survival rate, the Central Business District mainly consisted of commercial districts with reduced or stable survival rates in the future. In particular, most of the eastern regions were commercial districts with a declining survival rate in the future. The campus town area appeared to be a commercial area where the survival rate decreased in the past but increased or stable in the future. In the Gangnam Business District, various types of commercial districts were mixed. The main road is mainly a decreasing-maintenance commercial district. The west side of the main road showed a decrease in the future, and the east side of the main road showed a higher survival rate in the future. In addition, IT Business Districts were found to be commercial areas with an increased survival rate in the future.

Third, the types of commercial districts classified in this study are based on changes in the survival rate of existing businesses, which should be interpreted differently from the growth and decline of commercial districts. As a result of classifying the types of developed commercial districts and alley commercial districts in Seoul, the majority were decreased commercial districts, stable commercial districts, and increase after decrease commercial districts. The reasons for the decrease in the survival rate of existing businesses are: First, due to active entry of new businesses, competition between existing and new businesses increases the closure of existing businesses, or second, commercial districts decline, making it difficult to continue their operations. And the reasons for the increase after decrease in the survival rate of existing businesses are: First, the entry of new businesses decreases, weakening competition between existing and new businesses, or second, the growth of commercial districts makes existing businesses more active. In other words, when the commercial district grows, the survival rate of existing businesses may increase and decrease, which is the same even when the commercial district declines.

In addition, there is no relationship between this study's type of commercial district and the increase or decrease in sales. As a result of comparing the sales growth of each commercial district in 2014 and 2018 by commercial district type of developed commercial districts and alley commercial districts, the difference in sales growth by commercial district type was statistically insignificant. The reason is that the increase or decrease in the survival rate of existing businesses, which is based on the classification of commercial districts in this study, has a different meaning from the growth and decline of commercial districts.

As such, it is difficult to see that the change in the survival rate of existing businesses differs depending on the type of growth and decline in the commercial area. Therefore, to effectively establish the policy to prevent the closure of the self-employed business, it is necessary not only to consider the growth and decline of the commercial district but also to reflect the pattern of changes in the survival rate of existing businesses predicted in this study.

This paper is meaningful because it developed a more systematic and accurate prediction algorithm for the future survival rate of commercial facilities in Seoul. Based on this, we reclassified the commercial districts of Seoul in consideration of data and commercial characteristics. This study confirmed the possibility of using a deep learning-based predictive model for commercial district analysis. It is expected to be used as a basis for more direct and realistic policy establishment through accurate prediction of future business district changes in Seoul.

Nevertheless, this study has the following limitations. First, due to data limitations, variables such as floating popula-tion, resident population, and personal characteristics of business owners could not be applied to the survival rate pre-diction model. Second, since the future commercial district pattern changes were predicted based on historical data, it was impossible to directly reflect external phenomena that had an unexpectedly significant impact, such as COVID-19, in the model. Additionally, changes in the small businesses' survival rates do not always immediately reflect market trends. For instance, during an economic crisis, government or local authorities may implement funding support or revitalization projects to delay widespread closures, resulting in a lag in the reflection of broader economic or social trends. Third, even though k-means Clustering was performed twice, the range of the cluster is still vast, so some parts cannot be regarded as clearly distinguishing the types of commercial districts.

## Supporting information

**S1 Fig. The number of small business opening/closing and operating periods by industry.**
(TIF)

## Author contributions

**Conceptualization:** DongHyeon Lee, Jaekyung Lee, SangHyun Cheon.

**Data curation:** DongHyeon Lee, Jaekyung Lee, ManSu Kang, SangHyun Cheon.

**Formal analysis:** DongHyeon Lee, Jaekyung Lee, ManSu Kang, SangHyun Cheon.

**Funding acquisition:** SangHyun Cheon.

**Investigation:** DongHyeon Lee, Jaekyung Lee, ManSu Kang, SangHyun Cheon.

**Methodology:** DongHyeon Lee, Jaekyung Lee, ManSu Kang, SangHyun Cheon.

**Project administration:** DongHyeon Lee, Jaekyung Lee, ManSu Kang, SangHyun Cheon.

**Resources:** DongHyeon Lee, Jaekyung Lee, ManSu Kang, SangHyun Cheon.

**Software:** DongHyeon Lee, Jaekyung Lee, SangHyun Cheon.

**Supervision:** DongHyeon Lee, Jaekyung Lee, ManSu Kang, SangHyun Cheon.

**Validation:** DongHyeon Lee, Jaekyung Lee, SangHyun Cheon.

**Visualization:** DongHyeon Lee, Jaekyung Lee, SangHyun Cheon.

**Writing – original draft:** DongHyeon Lee, Jaekyung Lee, SangHyun Cheon.

**Writing – review & editing:** DongHyeon Lee, Jaekyung Lee, SangHyun Cheon.

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
