## [Decision Letter · Decision Letter 0]

Dear Dr. Lee,

Thank you for submitting your manuscript to PLOS ONE. After careful consideration, we feel that it has merit but does not fully meet PLOS ONE’s publication criteria as it currently stands. Therefore, we invite you to submit a revised version of the manuscript that addresses the points raised during the review process.

We look forward to receiving your revised manuscript.

Kind regards,

Guanghui Liu

Academic Editor

PLOS ONE

Journal Requirements:

4. We note that Figures 2, 5, 6, 7, 9 and 10  in your submission contain map/satellite images which may be copyrighted. All PLOS content is published under the Creative Commons Attribution License (CC BY 4.0), which means that the manuscript, images, and Supporting Information files will be freely available online, and any third party is permitted to access, download, copy, distribute, and use these materials in any way, even commercially, with proper attribution. For these reasons, we cannot publish previously copyrighted maps or satellite images created using proprietary data, such as Google software (Google Maps, Street View, and Earth). For more information, see our copyright guidelines: http://journals.plos.org/plosone/s/licenses-and-copyright.

a. You may seek permission from the original copyright holder of Figures 2, 5, 6, 7, 9 and 10  to publish the content specifically under the CC BY 4.0 license.  

Reviewers' comments:

Reviewer's Responses to Questions

**Comments to the Author**

1. Is the manuscript technically sound, and do the data support the conclusions?

Reviewer #1: Partly

Reviewer #2: Yes

Reviewer #3: Yes

2. Has the statistical analysis been performed appropriately and rigorously?

Reviewer #1: No

Reviewer #2: Yes

Reviewer #3: Yes

3. Have the authors made all data underlying the findings in their manuscript fully available?

Reviewer #1: Yes

Reviewer #2: Yes

Reviewer #3: No

4. Is the manuscript presented in an intelligible fashion and written in standard English?

Reviewer #1: Yes

Reviewer #2: Yes

Reviewer #3: Yes

Reviewer #1: To Authors,

Your research on the chronic closure of small businesses in South Korea addresses a critical economic issue with significant implications for policy and economic stability. The use of a large dataset of 3,000,000 individual commercial facility data spanning from 2004 to 2018 provides a robust foundation for your analysis. The dual focus of your study—developing a deep-learning based model to simulate survival rate patterns and reclassifying commercial districts based on prediction outcomes—is both innovative and highly relevant.

Your application of the Long Short-Term Memory (LSTM) model to predict and simulate the survival rate of commercial facilities is particularly noteworthy. The results indicating future decreases in survival rates in CBD areas, contrasted with increases in university districts and IT industry clusters, offer valuable insights. These findings not only enhance our understanding of commercial dynamics but also provide quantitative evidence that can inform more targeted and effective policy interventions. Below are some simple issues,

• Figure 1 fits in the appendix or this figure is not needed.

• More literature citations are needed. For example, lines 57 through 61 need to be supported by official reports or literature data.

• The “Year” in Table 3 means that the article is a cross-section?

• A note needs to be added below each image and table.

• The article is documented by year, so the major events mentioned at the end of the manuscript need to be analyzed as a split.

• The first letter of 4.1.2 needs to be capitalized.

• However, it would be beneficial to expand on the specific variables and features used in the LSTM model. Additionally, discussing the potential limitations of your model and the generalizability of your findings beyond Seoul could provide a more comprehensive view of your research.

Good Luck

Best,

Reviewer #2: The paper presented a very interesting topic analyzing the business survival rate in conjunction of the district wise business activity. It provided insightful conclusions regarding the business pattern within Seoul, and presented a generalizable framework and methodologies to gain valuable insights into local small businesses within major cities. Given the comprehensive techniques applied, well elaborated insights and good presentation, I recommend the paper to be published.

Meanwhile, here are my comments for authors to consider addressing:

1. Could we provide more information regarding data normalization and cleaning. The authors mentioned using industry wise data to create industry specific index and diversity index, is there any normalization performed between these industry level data, making sure that each industry data is at a similar level, with no outlier industry(s). The selection of the industries seem to have been provided in ref [23] and [28], it would be great if the authors could also provide the list of the industries used in the experiment within the paper for a full comprehensive understanding.

2. Authors created models based on constructed data within three time intervals, each of 5 years. Could we elaborate more regarding how the cutoff years were selected, and why did we choose to predict a 5 year survival rate instead of a different X year survival rate?

3. It would provide a more complete picture regarding how the LSTM model was trained, if we could provide examples of input data. LSTM is more superior in making predictions relying on data with sequential patterns, thus, a natural question would be, are we predicting survival rate using time series data or are we just using each independent variable combined? From here, providing examples of input data could provide clear insights into such questions.

4. Why did we choose to form the problem as a regression problem, since we are predicting a probability? What would be the argument between forming the problem as a classification, i.e. predicting survival or not, vs. forming the problem as a regression one? Is there any concern regarding the bounding values of predicted dependent variables, including probabilities larger than 1 or less than 0? If there is such concern, have we performed steps to normalize the outputs?

In all, I recommend the paper to be published given its novelty, insightful data and presentation, along with a generalizable framework for understanding urban businesses. It would be great if authors could address the comments and questions mentioned above.

Reviewer #3: The manuscript presents a valuable attempt to classify commercial districts in Seoul based on the survival rates of food service businesses using an LSTM-based prediction model. However, the paper requires revisions in methodology. Below are my detailed comments and suggestions for improvement.

Major issues:

1.While the study addresses an important issue, the novelty of the approach is questionable. The use of LSTM models for time-series prediction is well-established, and the application to commercial districts, while interesting, does not seem to provide significant new insights or methods.

2. The manuscript lists seven independent variables used in the LSTM model. However, the rationale for selecting these specific variables is insufficiently explained. While the literature is cited, it is not clear why these particular variables were chosen over others. Additionally, the manuscript should discuss whether other potentially relevant variables were considered and why they were excluded. For example, factors such as local economic indicators, consumer behavior data, or external shocks (e.g., economic crises, pandemics) could significantly impact survival rates.

3. It would be beneficial to include a discussion comparing LSTM with other potential models, such as AR, MA, ARIMA, GRU (Gated Recurrent Unit).

4. Given that the classification of commercial districts can significantly influence the results and interpretations, it would be beneficial to conduct a sensitivity analysis to determine how changes in the classification thresholds might affect the outcomes. For instance, what happens if the thresholds are adjusted to mean ±1 standard deviation? Does the distribution of commercial districts change significantly, and how does this impact the subsequent analysis? Including a sensitivity analysis would provide more confidence in the robustness of the classification method.

5. The manuscript provides a general description of how different business densities are distributed across Seoul (e.g., high-density districts are major commercial areas). However, the spatial distribution of these density classifications is not deeply analyzed in relation to the urban geography of Seoul. The manuscript should explore how these classifications relate to existing urban structures, such as zoning regulations, transportation networks, and residential patterns. For example, are high-density commercial districts primarily located near major transportation hubs or within specific urban zones? Understanding these spatial relationships would add depth to the analysis and provide more meaningful insights into urban planning.

6. The manuscript does not address whether the classification of business densities is stable over time or if there are significant temporal variations. For instance, a district classified as medium-density in one time period might transition to high-density in another period due to urban development or economic shifts.

7. Survival rate dynamics capture changes over time, but the manuscript does not discuss the potential for lag effects, where changes in survival rates may not immediately reflect broader economic or social trends. For instance, a commercial district might experience a decline in survival rates due to a delayed response to economic downturns or shifts in consumer behavior.

Minor issues:

1. Were the data normalized or standardized?

2. Is there any missing data? How were missing values handled? Any imputation methods used?

**Do you want your identity to be public for this peer review?** For information about this choice, including consent withdrawal, please see our Privacy Policy

Reviewer #1: No

Reviewer #2: No

Reviewer #3: **Yes: ** Wenhao Li

---

## [Author Response · Author response to Decision Letter 1]

25 Feb 2025

Thank you for your valuable comments. The recommendations and suggestions have significantly improved the quality of this paper.

For detailed responses, please refer to the submitted file, *Response to Reviewers*.

---

## [Decision Letter · Decision Letter 1]

Dear Dr. Lee,

Thank you for submitting your manuscript to PLOS ONE. After careful consideration, we feel that it has merit but does not fully meet PLOS ONE’s publication criteria as it currently stands. Therefore, we invite you to submit a revised version of the manuscript that addresses the points raised during the review process.

We look forward to receiving your revised manuscript.

Kind regards,

Guanghui Liu

Academic Editor

PLOS ONE

**Journal Requirements:**

Reviewers' comments:

Reviewer's Responses to Questions

**Comments to the Author**

Reviewer #1: All comments have been addressed

Reviewer #2: All comments have been addressed

Reviewer #3: (No Response)

2. Is the manuscript technically sound, and do the data support the conclusions?

Reviewer #1: Yes

Reviewer #2: Yes

Reviewer #3: Yes

3. Has the statistical analysis been performed appropriately and rigorously?

Reviewer #1: Yes

Reviewer #2: Yes

Reviewer #3: Yes

4. Have the authors made all data underlying the findings in their manuscript fully available?

Reviewer #1: Yes

Reviewer #2: Yes

Reviewer #3: No

5. Is the manuscript presented in an intelligible fashion and written in standard English?

Reviewer #1: Yes

Reviewer #2: Yes

Reviewer #3: Yes

**Reviewer #1:**  Thank you for your hard work. From the perspective of the methodology, I think you have done a very good job.

**Reviewer #2: ** Given the comprehensiveness, techniques, and detailed description of the conducted research, I would recommend the paper to be published (after addressing 2 very minor items identified below)

The previously left comments by me were well addressed:

1. Authors added a more in-depth explanation regarding the preparation of the data, provided more information regarding how were industry-specific index and diversity index selected and prepared.

2. Authors now provided a paragraph describing difference between LSTM and other time series and sequential models. Information of how the data was split for training and validation is also provided.

3. Time intervals were clearly marked out, and authors did a good job explaining why 5 years were selected as a training and prediction interval.

4. Authors explained why the problem was treated as a regression instead of classification.

5. Authors provided more information in the rebuttal letter providing detailed information regarding data availability and addressing copyright questions.

In all, I'd recommend the paper to be accepted for publication, however, there are 2 very minor items I'd like authors to quickly address:

1. In the paragraph that describing the data/features used in the modeling, authors mentioned that they were relying on information based from "One hundred industries closely tied to daily life", however, I'm not sure if I'm seeing corresponding reference here, could we please add this information.

2. I'm not sure if I missed it, but I don't think I'm seeing Figure 3. LSTM conceptual diagram in both the main content nor supplement materials.

**Reviewer #3: ** I have two minor comments:

1) given the dynamic nature of commercial districts, it would be interesting to discuss the potential for using the LSTM model for dynamic prediction or real-time predictions. For example, could the model be integrated into a dashboard that provides up-to-date survival rate predictions for different districts?

(Taylor JM, Yu M, Sandler HM. Individualized Predictions of Disease Progression Following Radiation Therapy for Prostate Cancer. Journal of Clinical Oncology 2005; 23(4): 816–825.)

2) Has cross validation been used for prediction accuracy?

**Do you want your identity to be public for this peer review?** For information about this choice, including consent withdrawal, please see our Privacy Policy

Reviewer #1: No

Reviewer #2: No

Reviewer #3: No

---

## [Author Response · Author response to Decision Letter 2]

28 Apr 2025

Thank you for your valuable comments.

The recommendations and feedback provided have been very helpful in improving the quality and clarity of our manuscript.

In response, we have carefully revised the manuscript to address all comments.

A detailed, point-by-point response has been included in the uploaded response document.

Thank you again for your thoughtful review.

---

## [Decision Letter · Decision Letter 2]

Classification of commercial districts based on predicting the survival rate of food service market in Seoul

PONE-D-24-30481R2

Dear Dr. Lee,

We’re pleased to inform you that your manuscript has been judged scientifically suitable for publication and will be formally accepted for publication once it meets all outstanding technical requirements.

Kind regards,

Guanghui Liu

Academic Editor

PLOS ONE

Additional Editor Comments (optional):

Reviewers' comments:

Reviewer's Responses to Questions

**Comments to the Author**

Reviewer #3: All comments have been addressed

2. Is the manuscript technically sound, and do the data support the conclusions?

Reviewer #3: Yes

3. Has the statistical analysis been performed appropriately and rigorously?

Reviewer #3: Yes

4. Have the authors made all data underlying the findings in their manuscript fully available?

Reviewer #3: Yes

5. Is the manuscript presented in an intelligible fashion and written in standard English?

Reviewer #3: Yes

Reviewer #3: All of my previous comments have been adequately addressed in the revised manuscript. I believe the authors have made the necessary improvements, and I recommend the manuscript for publication.

**Do you want your identity to be public for this peer review?** For information about this choice, including consent withdrawal, please see our Privacy Policy

Reviewer #3: **Yes: ** Wenhao Li

---

## [Editor Report · Acceptance letter]

PONE-D-24-30481R2

PLOS ONE

Dear Dr. Lee,

I'm pleased to inform you that your manuscript has been deemed suitable for publication in PLOS ONE. Congratulations! Your manuscript is now being handed over to our production team.

Kind regards,

on behalf of

Dr. Guanghui Liu

Academic Editor

PLOS ONE